# Anti-neuraminidase and anti-HA stalk antibodies reduce the susceptibility to and infectivity of influenza A/H3N2 virus

Gregory Hoy[1,11], Thomas Cortier[2,3,11], Hannah E. Maier [1], Guillermina Kuan[4,5], Roger Lopez[4,6], Nery Sanchez[4], Sergio Ojeda[4], Miguel Plazaola[4], Daniel Stadlbauer[7], Abigail Shotwell[1], Angel Balmaseda[4,6], Florian Krammer [7,8,9,10], Simon Cauchemez [2,12] ✉ & Aubree Gordon [1,12] ✉

Immune responses against neuraminidase (NA) and hemagglutinin (HA) are critical for developing effective influenza vaccines, yet their role in influenza transmission remains unclear. We conducted household transmission studies in Managua, Nicaragua, to assess the impact of anti-NA and anti-HA antibodies induced by natural infection on influenza A/H3N2 susceptibility and infectivity. Using mathematical models capturing household transmission dynamics, we found that high pre-existing antibody levels against the HA head (>31, [95% CrI 13–67]), HA stalk (>35, [95% CrI 11–66]), and NA (>31, [95% CrI 12–68]) are associated with reduced susceptibility to infection (relative susceptibility: HA head, 0.63 [95% CrI 0.42–0.98]; HA stalk, 0.66 [95% CrI 0.44–0.99]; NA, 0.49 [95% CrI 0.30–0.76]). HA stalk (>58 [95% CrI: 47–70]) and NA (>27 [95% CrI: 15–43]) are associated with reduced infectivity (relative infectivity: NA, 0.55 [95% CrI: 0.32–0.98], HA stalk 0.53 [95% CrI: 0.27–0.97]). These findings suggest that influenza vaccines designed to elicit NA immunity in addition to HA immunity may not only enhance protection against infection but also reduce onward transmission.

Influenza virus infection remains an important cause of human disease burden, with upwards of one billion infections and up to 650,000 deaths occurring globally every year (https://www.who.int/news/item/11-03-2019-who-launches-new-global-influenza-strategy). Vaccination against influenza virus is one of the most effective approaches for reducing the overall morbidity and mortality of seasonal influenza in communities, and improving the effectiveness of influenza vaccines is

an important goal[1–4]. There are two important components of transmission. Susceptibility refers to an individual or group's propensity to become infected with influenza, assuming adequate exposure. Infectivity refers to an individual or group's propensity to infect others, assuming that they themselves are infected (Fig. 1). Conditional on the first person already being infected, the overall risk of transmission from one person to another depends on the infectivity of the first

[1]School of Public Health, University of Michigan, Ann Arbor, MI, USA. [2]Mathematical Modelling of Infectious Diseases Unit, Institut Pasteur, Université Paris Cité INSERM U1332, CNRS UMR2000 Paris, France. [3]Collège Doctoral, Sorbonne Université, Paris, France. [4]Sustainable Sciences Institute, Managua, Nicaragua. [5]Centro de Salud Sócrates Flores Vivas, Ministry of Health, Managua, Nicaragua. [6]Laboratorio Nacional de Virología, Centro Nacional de Diagnóstico y Referencia, Ministry of Health, Managua, Nicaragua. [7]Department of Microbiology, Icahn School of Medicine at Mount Sinai, Icahn School of Medicine at Mount Sinai, New York, NY, USA. [8]Center for Vaccine Research and Pandemic Preparedness (C-VaRPP), Icahn School of Medicine at Mount Sinai, New York, NY, USA. [9]Department of Pathology, Molecular and Cell-Based Medicine, Icahn School of Medicine at Mount Sinai, New York, NY, USA. [10]Ignaz Semmelweis Institute, Interuniversity Institute for Infection Research, Medical University of Vienna, Vienna, Austria. [11]These authors contributed equally: Gregory Hoy, Thomas Cortier. [12]These authors jointly supervised this work. Simon Cauchemez, Aubree Gordon. ✉e-mail: simon.cauchemez@pasteur.fr; gordonal@umich.edu

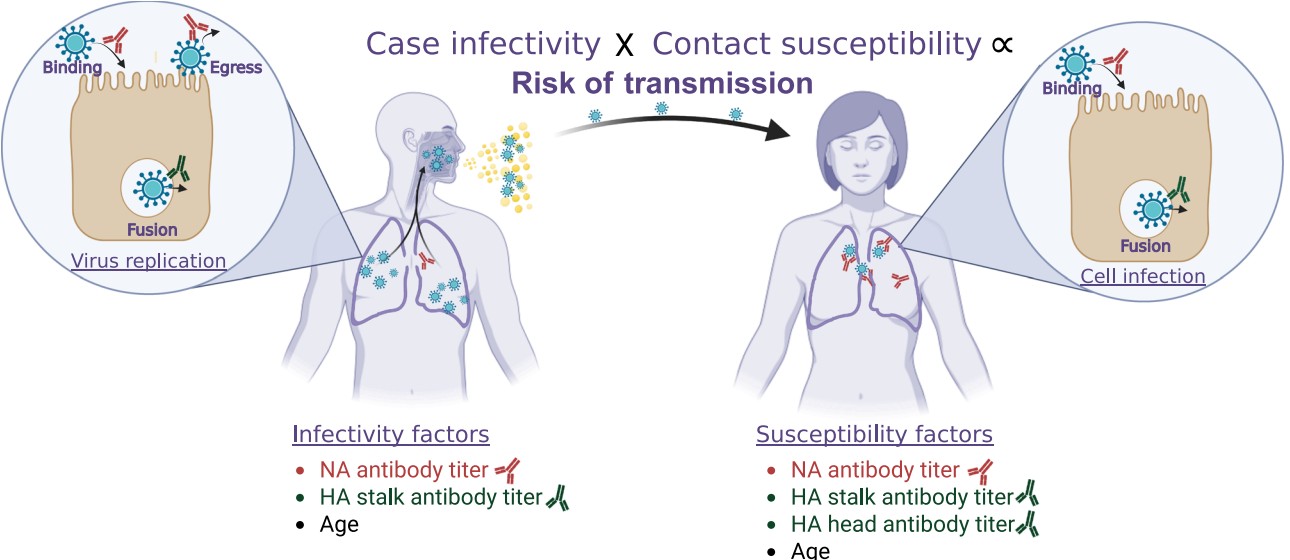

**Fig. 1 | Factors influencing pairwise transmission risk, as tested across the different models used in the study.** Zoomed bubbles strands for the potential cellular mechanism of antibodies on modulation of infectivity (left) and susceptibility (right). NA refers to neuraminidase, and HA refers to hemagglutinin. Created in BioRender. Cortier, T. (2025) https://BioRender.com/tlkiwj8.

person and the susceptibility of the second. Much of the effort for influenza vaccine improvement has focused on the induction of immune responses that reduce susceptibility to infection or disease, to moderate effectiveness. Much less attention has been given to the development of vaccines that reduce individual-level infectivity among the vaccinated; in other words, vaccines that generate an immune response that decreases the infectivity of vaccinees, even if they are not fully protected from infection (https://www.cdc.gov/flu/vaccines-work/past-seasons-estimates.html)[2,3,5,6]. Population-level vaccination efforts clearly reduce overall community transmission, likely because vaccinated individuals show reduced susceptibility to infection, which breaks transmission chains[7–9]. However, there is no evidence that current-generation influenza vaccines reduce individual-level infectivity directly, and little is known about if and how immune responses that protect against influenza virus infection affect individual infectivity. The identification of immune responses that both reduce susceptibility to infection and reduce infectivity among vaccinated individuals who are infected would allow for the development of influenza vaccinations that lower overall influenza circulation in communities and that provide additional indirect protection to individuals who are unvaccinated or under-vaccinated for influenza, including infants and immunocompromised individuals.

Antibody responses against neuraminidase (NA), an influenza surface glycoprotein, are thought to protect against severe disease caused by the influenza virus, and high pre-infection anti-NA antibody levels have been shown to reduce the overall duration of influenza viral shedding[10–17]. Additionally, studies in animal models have demonstrated a reduction in viral shedding occurring in animals immunized against NA[18,19]. Multiple studies have highlighted the essential role of the HA stalk in shaping viral shedding within animal models, such as mice[20], where HA function, modulated by pH control, impacted shedding, or by comparing different HA mutants in chickens[21], and even in healthy patients with preexisting high levels of HA stalk antibodies[22]. However, viral shedding does not always consistently correlate with infectivity, and direct transmission reduction of anti-NA immunity has not been demonstrated in human populations[23]. Additionally, the role of the anti-NA response on infectivity, relative to other important immune targets such as hemagglutinin (HA), has not been investigated. This study aims to explore the impact of pre-existing antibody levels against HA head, HA stalk, and NA on influenza virus A/H3N2 transmission in a household

setting, with particular interest in the role of anti-neuraminidase responses in modulating transmission risk.

## Results
### Participant and household characteristics
Over three influenza seasons (2014, 2016, 2017), a total of 171 households (171 index cases and their 664 household contacts) were recruited following the detection of an infected individual (i.e., the index case) and followed up for an average duration of 36.7 days. Households were enrolled through identification of an index case at the study clinic (2014, 2016), or pre-enrolled households were activated after detection of influenza virus via polymerase chain reaction (PCR) in a member of an enrolled household (2017). Once activated, index cases and household contacts were tested for influenza virus every 2–3 days using PCR, and serology was done on blood samples collected on the first day of household activation (the initial/acute sample) and 30–45 days after household activation (the final/convalescent sample). More information about the study design and case ascertainment is available in the Online Methods. 148 out of 664 (22.3%) household contacts were infected during the follow-up period, 13% in 2014, 26% in 2016, and 23% in 2017 (Fig. 2). The number of infections per household ranged from 1 to 10, with an average of 0.87 secondary infections per household. The serial interval (i.e., the average time between index case onset and onset of cases in household contacts) was 3.4 days (SD 2.8 days). A visualization of the intensive monitoring periods by household is presented in Fig. 2.

Less than 10% of participants had ever been vaccinated against influenza, and only two individuals had been vaccinated for influenza in the 6 months prior to the start of the monitoring period. Influenza A/H3N2 positive individuals were younger. Overall, all pre-existing antibody levels against the HA head, HA stalk, and NA were lower in infected individuals than in uninfected individuals ($p$-values $< 10^{-5}$; Table 1 and Fig. 2). HA stalk and NA antibody levels were significantly lower in children than in adults ($p < 10^{-5}$; Fig. 2), reflecting their lower exposure to prior influenza epidemics. The HA head titer was higher in children compared to adults ($p = 1.4 \times 10^{-4}$).

### Modeling the probability of secondary infection
We used boosted regression trees to investigate the impact of antibody titers and age of the index case and their household contacts on

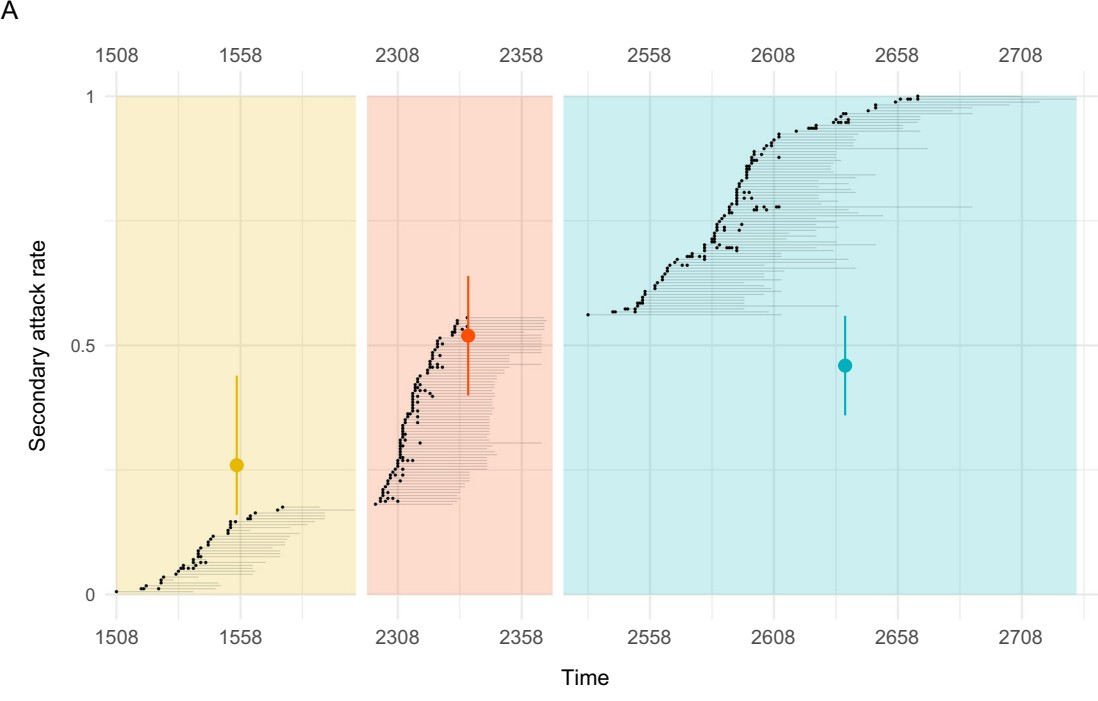

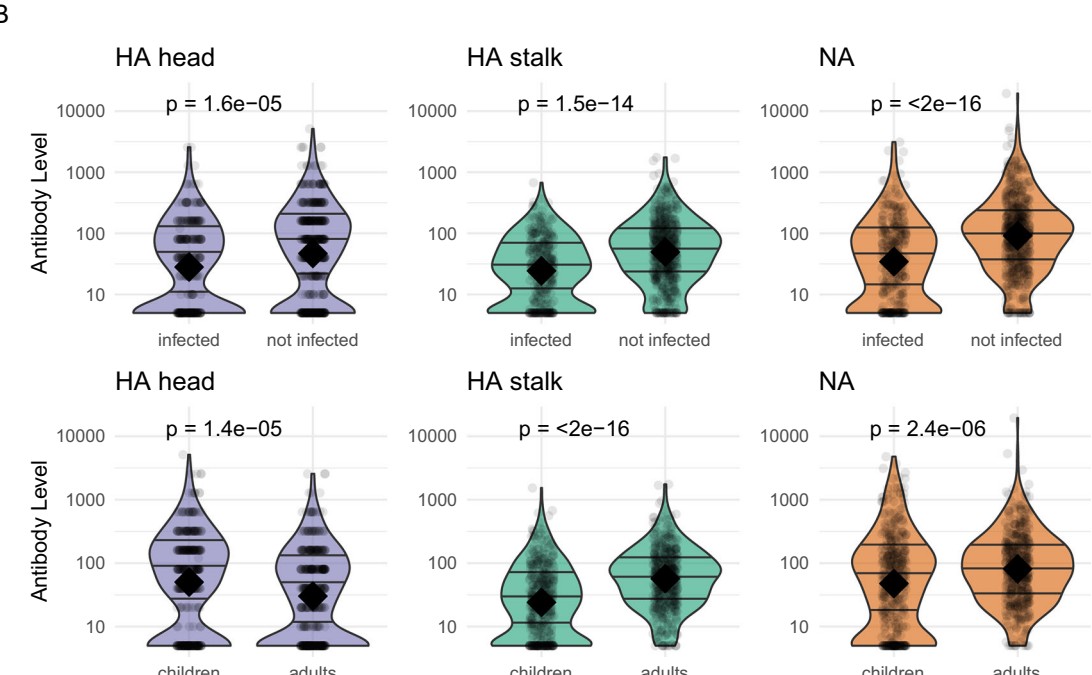

**Fig. 2 | Household epidemiological and antibody data. A** Follow-up periods and infection events of 171 households. Gray segments represent the follow-up periods for each household. Black dots indicate the dates of the first positive PCR test for household cases. The background colors correspond to the three waves studied. Colored dots with error bars (95% Confidence interval) show the secondary attack rate (SAR) during each wave. **B** Distribution of pre-infection antibody titers for PCR-negative (not infected) and PCR-positive (infected) individuals is shown in the top row, and stratified by age in the bottom row. Black dots represent individual data points. The black diamond indicates the mean titer for each group, while the three horizontal lines represent the 25th, 50th (median), and 75th percentiles. HA head refers to the hemagglutinin head, in light blue. HA stalk refers to the hemagglutinin stalk, in green. NA refers to neuraminidase, in orange. Mann–Whitney $U$ tests were performed to compare antibody distributions.

the probability of secondary infection (Fig. 3). Unlike traditional regression models, this approach captures non-linear relationships and complex interactions between variables, allowing for the estimation of diverse functional relationships. We found that the probability of secondary infection was significantly associated with the ages of the household contacts ($p = 0.02$) and of the index case ($p < 0.01$). This probability was also significantly influenced by the NA, HA stalk, and

HA head antibody titers of the household contact (all $p < 0.01$) as well as the NA and HA stalk antibody titers of the index case (both $p = 0.01$).

Figure 3 shows the associations of each predictor covariate (age and antibody levels) with the probability of becoming infected (Fig. 3A–D) and the probability of infecting others (Fig. 3E–H). Covariate importance was quantified using mean absolute SHAP values, which measure the average contribution of each covariate to the

**Table 1 | Descriptive statistics of study population**

|  | A/H3N2 positive (n = 319) n (%) median (sd) | A/H3N2 negative (n = 516) n (%) median (sd) | Total (n = 835) n (%) median (sd) | p-value |
|---|---|---|---|---|
| Gender |  |  |  |  |
| Female | 176 (55.2) | 342 (66.3) | 518 (62.0) | 0.0016 |
| Male | 143 (44.8) | 174 (33.7) | 317 (38.0) |  |
| Age | 9 (14.9) | 23 (19.0) | 15 (18.6) | <2e-16 |
| Vaccination |  |  |  |  |
| Ever vaccinated | 40 (12.5) | 36 (7.0) | 76 (9.1) | 0.009 |
| Recently Vaccinated | 2 (0.6) | 0 (0.0) | 2 (0.2) | NA |
| Season |  |  |  |  |
| 2014–2015 | 44 (13.8) | 90 (17.4) | 134 (16.0) | 0.1135 |
| 2016–2017 | 122 (38.2) | 164 (31.8) | 286 (34.3) |  |
| 2017–2018 | 153 (48.0) | 262 (50.8) | 415 (49.7) |  |
| HA stalk | 26 (68) | 55 (168) | 40 (140) | 1e-15 |
| HA head | 40 (246) | 80 (387) | 40 (341) | 2e-5 |
| NA | 36 (284) | 98 (970) | 70 (785) | <2e-16 |

p-values were obtained with chi-squared test of independence for categorical variables and by two-sided Mann–Whitney U tests for continuous variables.

model's predictions (see "Methods"). This revealed that the strongest impacts on transmission were due to household contact HA stalk antibody titer (SHAP = 0.36) and NA titer (SHAP = 0.29), followed by index case HA stalk titer (SHAP = 0.25), age (SHAP = 0.24), NA titer (SHAP = 0.23), and finally household contact age (SHAP = 0.23) and HA head titer (SHAP = 0.20).

Index cases with low levels of NA titers (Fig. 3F) had a high probability of infecting others, but the probability decreased rapidly with increasing NA titers. In contrast, household contacts' susceptibility and index cases' infectivity decreased more gradually with increasing HA stalk antibody levels (Fig. 3C–G). Susceptibility of household contacts was high for low titers against the HA head and NA (below 30) (Fig. 3B, D).

### Effect of pre-existing antibody levels on susceptibility and infectivity

A mathematical model describing household transmission dynamics was calibrated to the data with Bayesian data augmentation methods[24], to account for different possible routes of transmission, including tertiary infections (i.e., household member infected by another household member who is not the index case), and from possible reintroduction from the community (i.e., household member infected outside the household), estimate household transmission risks and determine factors affecting individual relative susceptibility and infectivity (see Online "Methods")[25]. With this approach, we estimate both the reduction in susceptibility and infectivity associated with high titers compared to low titers, and the optimal titer thresholds to explain the differential ability of individuals to become infected or transmit influenza. To facilitate the estimation of the optimal threshold value, we provided a prior informed by the analysis presented in Fig. 3 (see "Methods").

Pre-existing antibody levels were associated with reduced susceptibility. Specifically, antibody titers against the HA head above 31 (95% CrI: 13–67) reduced susceptibility by 0.64 (95% CrI: 0.42–0.98), while titers against the HA stalk above 35 (95% CrI: 11–67) reduced susceptibility by 0.66 (95% CrI: 0.44–0.99). Similarly, titers against NA above 31 (95% CrI: 12–67) reduced susceptibility by 0.49 (95% CrI: 0.30–0.76) (Fig. 4A). Compared to adults aged 15+ years, children aged 0–14 years exhibited higher relative susceptibility (2.53, 95% CrI: 1.67–3.82) (Fig. 4F).

Among individuals infected with A/H3N2, pre-existing antibody levels were associated with reduced infectivity. NA titers above 27 (95% CrI: 15–43) were linked to a relative infectivity of 0.55 (95% CrI: 0.32–0.98), while HA stalk titers above 58 (95% CrI: 47–70) were associated with a relative infectivity of 0.53 (95% CrI: 0.27–0.97) (Fig. 5A–D). In contrast, high antibody levels against the HA head were not associated with reduced infectivity, regardless of the threshold used (Supplementary Fig. 1). Children also exhibited lower infectivity than adults, with a relative infectivity of 0.58 (95% CrI: 0.34–0.98) (Fig. 5E). Although the secondary attack rate (the proportion of household contacts that become infected) appeared lower during the first wave, this finding was non-significant (RR = 0.71; 95% CrI: 0.41–1.21). Consistent with previous household transmission studies, transmission risk decreased with household size; for example, households of size 4 had 69% (95% CrI: 51–91) relative transmission compared to households of size two (Supplementary Fig. 2). The probability of infection from the community was estimated at 0.04% per day, reflecting the fact that most households were recruited during epidemic waves.

### Simulation studies and sensitivity analyses

Simulating epidemics in households using our model, we found that the transmission model parameters accurately captured the observed patterns of secondary attack rates (SARs) across household sizes (Supplementary Fig. 4).

To validate our inference framework, we applied it to data simulated from our model with known parameter values. The parameter estimates were consistently recovered without directional bias (Supplementary Fig. 5 6), and the true simulation values fell within the 95% credible interval in more than 94% of simulations for all parameters (Supplementary Table 4). In Supplementary Fig. 3, we validate the assumption that the effects of different types of antibodies on susceptibility and infectivity are additive.

Overall, parameter estimates were robust to prior assumptions. A wider prior on relative risks (log-SD = 2 vs. 1) preserved both point estimates and significance, suggesting that the default prior (log-SD = 1) was not overly restrictive. In the baseline scenario, the prior ranges for antibody activation thresholds were informed by boosted regression tree results (Fig. 3). Sensitivity analyses tested uniform priors of varying widths (20, 40, 60, 80) centered around the target threshold values for both susceptibility and infectivity. Point estimates for the reduction in susceptibility and infectivity remained stable across all tested conditions. However, with the widest prior ranges (60 and 80), the relative risk estimates associated with NA and HA stalk were less precise, with credible intervals overlapping 1, suggesting borderline significance. We present a scenario with an informed prior in the main for the sake of clarity.

Point estimates remained stable for a variety of assumptions about the incubation period (from 0.6 days to 2 days) and infectivity duration (3 to 4 days) distributions (Supplementary Table 1.3). When the mean incubation period increased from 1 to 2 days, some borderline significant parameters—such as the effect of HA stalk on susceptibility and NA on infectivity—shifted slightly above 1 in the upper value of the 95% credible intervals. Similarly, reducing the mean generation time to 3 days (from 3.5) led to the upper bound for the effect of HA stalk on infectivity exceeding 1. In contrast, increasing the generation time to 4 days or reducing the incubation period to 0.6 days preserved all significant results relative to the baseline scenario.

### Discussion

Through intensive monitoring of households with confirmed influenza A/H3N2 infections and statistical transmission modeling, this study assessed the impact of three individual broadly neutralizing antibody candidates—relevant to next-generation vaccines—on susceptibility and infectivity within the household setting.

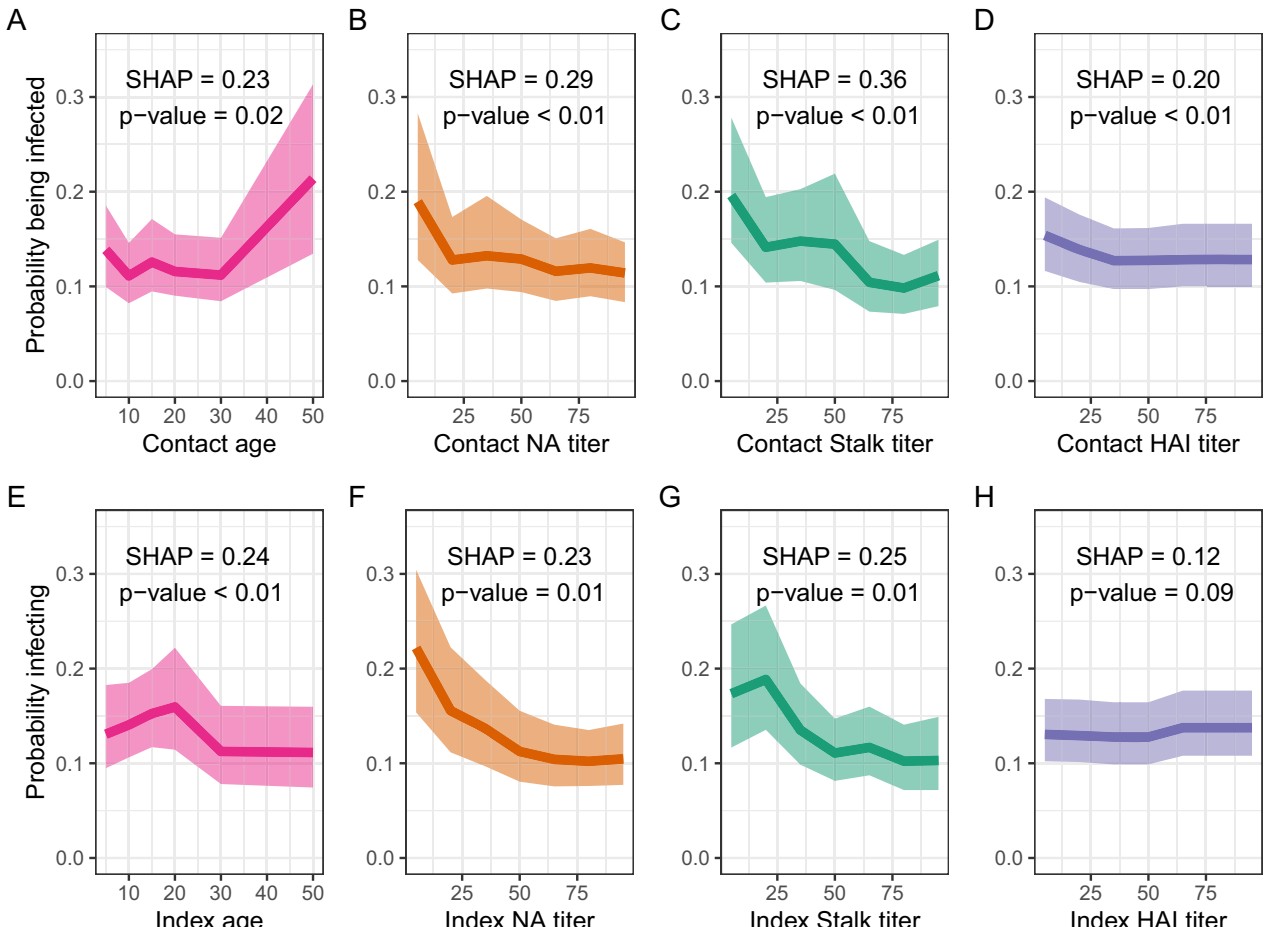

**Fig. 3 | Association between age, antibody titers of the household index case, antibody titers of contact individuals, and the probability of infection in contact individuals, from boosted tree regression.** The first row (**A**, **B**, **C**, **D**) shows the effect of contact's individual factors (e.g., age and antibody titers) on the probability of secondary infection. The second row (**E**, **F**, **G**, **H**) shows the effect of index case's factors (e.g., age and antibody titers) on their probability of becoming infected. *p*-values were obtained using permutation tests (see methods for details). SHAP refers to the SHapley Additive exPlanations. Pink designates age; orange designates NA (neuraminidase); green designates hemagglutinin stalk; purple designates HAI (hemagglutinin inhibition assay). The shaded area represents the 95% CrI of posterior predictions from the 500 boosted regression tree models.

Individuals infected with influenza A/H3N2 who had high pre-existing antibody levels against NA or HA stalk exhibited significantly reduced infectivity compared to those with low preexisting antibody levels. Importantly, preexisting antibodies against the HA head were not associated with reduced infectivity, indicating that a reduction in influenza infectivity may depend on anti-NA and anti-HA stalk responses alone. This specificity is biologically plausible, as NA and HA are implicated in viral egress[26,27], and thus it is reasonable that anti-NA and anti-HA stalk responses contribute to a reduction in overall viral load, viral shedding, and subsequent transmission[14]. Antigenic drift in the HA head could lead to worsening correlation between anti-HA head antibodies, measured using HA head with the Hong Kong 2014 antigen, and protection against infection, and subsequently an under-representation of the protective benefit of anti-HA head responses relative to anti-HA stalk and anti-NA responses. Though a theoretical explanation for the findings of this study, an analysis with a very similar analytic subset demonstrated that the correlation between anti-HA head antibodies and protection against infection was not improved with the use of better-matching anti-HA antigens, such as Singapore 2016[28].

We found that age and pre-existing antibody levels against HA head, HA stalk, and NA, were associated with influenza susceptibility. Individuals with high preexisting antibodies demonstrated reduced susceptibility to influenza A/H3N2 virus infection, with reductions of 37%, 34%, and 42%, respectively, for antibodies against the HA head, HA stalk, and NA, which is in line with previous work on the correlates of protection against influenza viruses, including influenza A/H3N2[10,11,29,30]. Children aged 14 years and younger were more than two times more susceptible than adults aged 15 years and older. This is in line with a large body of work indicating that children are more susceptible to influenza virus infection but suggests that this susceptibility is not entirely due to a lack of immune response against antigens that we measure, as we observe a large association between age and susceptibility even when accounting for lower anti-HA stalk and anti-NA antibody levels among children[23,31–35].

The independent effects of the three tested antibodies on both susceptibility and infectivity were supported by a supplementary analysis evaluating all possible antibody combinations, which produced results consistent with our baseline model. This is consistent with the fact that different parts of the viral replication cycle are targeted by these antibodies.

We highlight the potential value of inducing strong anti-NA and anti-HA stalk immunity as key components of future influenza vaccine strategies. Though current-generation influenza vaccines typically include a neuraminidase component, the immunogenicity of the NA component of vaccines is inconsistent, and anti-NA responses are

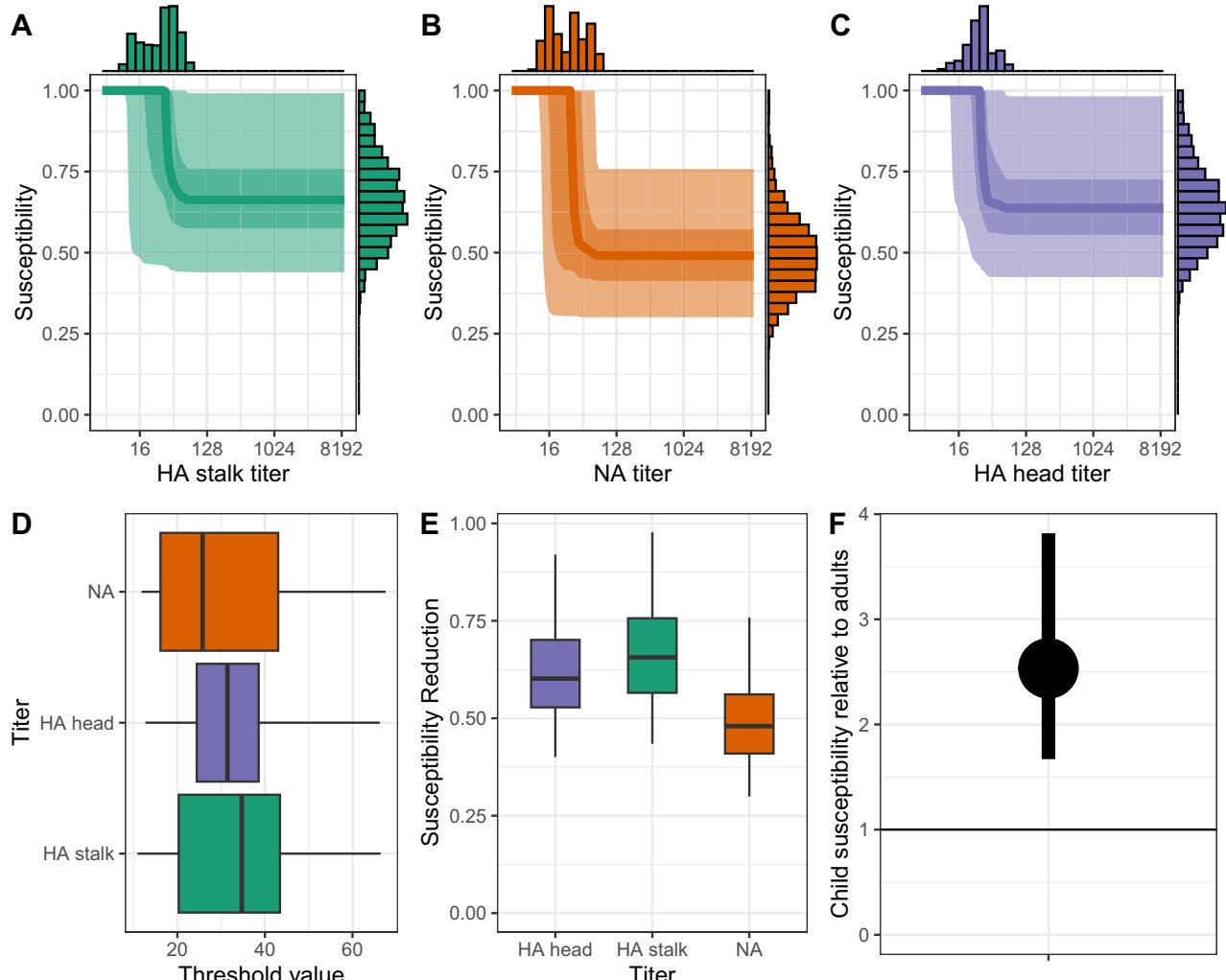

**Fig. 4 | Impact of pre-existing antibodies and age on susceptibility to infection.**
**A** Estimated association between HA stalk titer and susceptibility reduction. The solid line represents the median association, the darker shaded area indicates the 50% credible interval (CI), and the lighter shaded area shows the 95% CI. The top and side histograms illustrate the posterior distributions of the threshold values and susceptibility reduction, respectively. **B** Estimated association between NA titer and susceptibility reduction, with median and credible intervals displayed as in (**A**). **C** Estimated association between HA head titer and susceptibility reduction, with median and credible intervals displayed as in (**A**). **D** Comparison of the posterior distributions of the threshold values for the three antibody titers (HA stalk, NA, and HA head). **E** Comparison of the posterior distributions of the susceptibility reduction for the three antibody titers (HA stalk, NA, and HA head). **F** Relative susceptibility of children aged ≤15 years compared to adults aged >15 years (median and 95% CrI). The boxplots represent these five percentiles (2.5%, 25%, 50%, 75%, and 97.5%) of the posterior distributions.

often not utilized as an endpoint in the projection of vaccine efficacy against seasonal influenza[4,36]. These results suggest that, to generate next-generation influenza vaccines that are effective at reducing susceptibility as well as infectivity, the anti-NA response generated by vaccine candidates needs to be emphasized, measured, and assessed.

The secondary attack rate in the study population was 22.3%, which is consistent with that found from other studies, especially given that there is a large proportion of children who have been associated with higher influenza SARs, in this population relative to that of other studies[35,37].

Furthermore, the robustness of the transmission model results was reinforced by the findings of the boosted regression trees, which explained the infectious status of contact individuals based on the antibody levels of the index case. Our estimates indicate that higher anti-HA stalk and anti-NA antibody levels in the index case significantly reduced the probability of transmission within households.

This study is strengthened by the relatively large sample size, robust immunologic characterization of participants before and after

infection, and methods that allow for probabilistic chains of transmission to be reconstructed to assess individual-level risk factors for infectivity and susceptibility, rather than assuming all secondary cases arise from the index case. Furthermore, the unvaccinated nature of the population allows us to specifically examine infection-induced immunity.

This study is limited by the low number of vaccinated participants that makes stratification by vaccination status impossible, and the relative contribution of anti-NA antibodies on A/H3N2 transmission may differ between vaccine-induced and infection-induced immunity in ways that we cannot assess. Furthermore, even though our modeling accounts for the possibility of community infections, we cannot rule out the possibility that some household contacts infected in the community might have been misattributed to in-household transmission; however, we would not expect this to be specific to higher or low-to-undetectable antibody levels for any of the targets, and would not expect a directional bias in these estimates. The model assumes that all individuals share the same transmission risk profile over time,

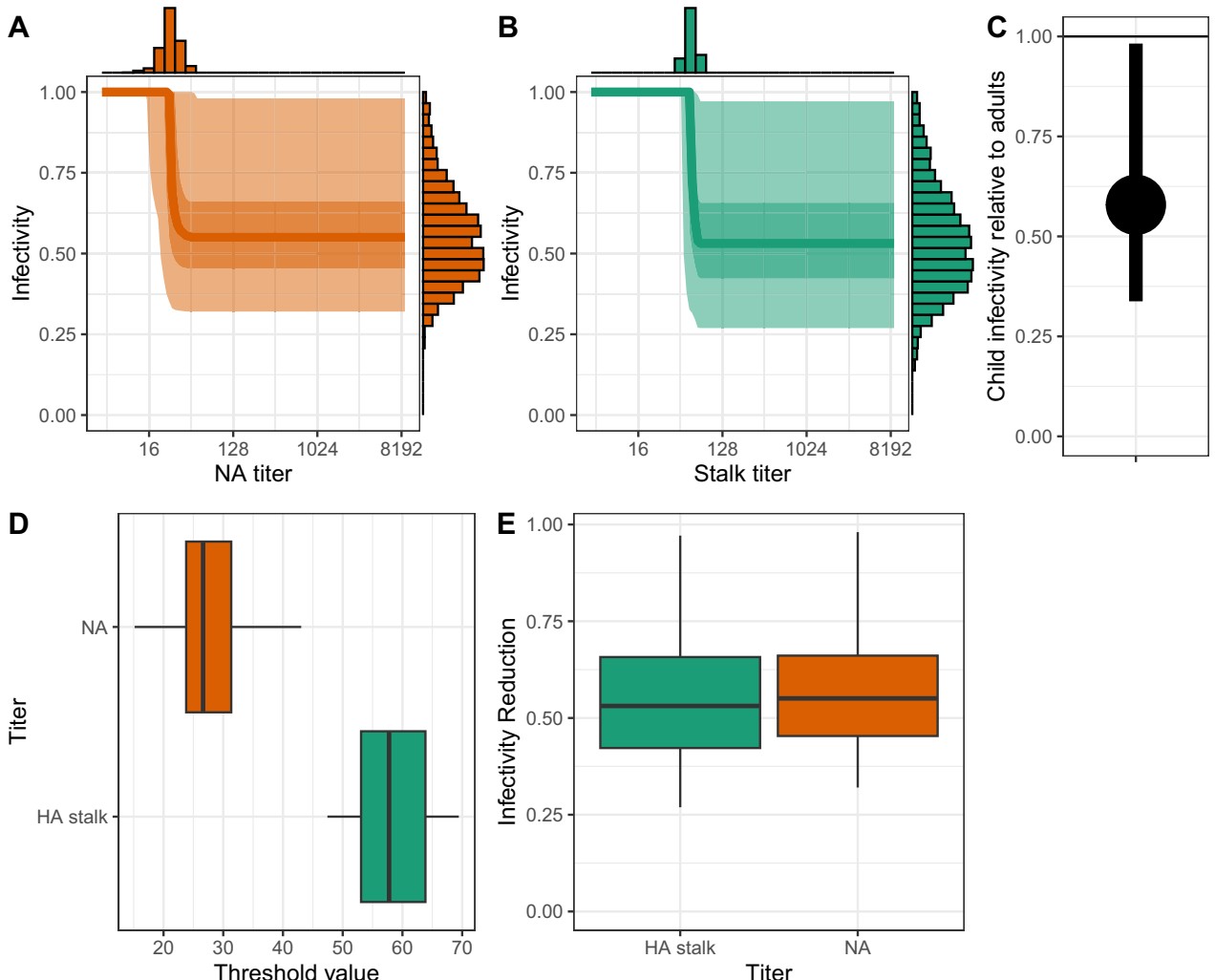

**Fig. 5 | Impact of preexisting antibody titers and age on infectivity. A** Estimated association between NA titer and infectivity. The solid line represents the median association, the darker shaded area denotes the 50% credible interval (CI), and the lighter shaded area shows the 95% CI. Top and side histograms display the posterior distributions of threshold values and infectivity reduction, respectively. **B** Estimated association between HA stalk titer and infectivity, with median and credible intervals displayed as in (**A**). **C** Comparison of the posterior distributions of the threshold values for NA and HA stalk titers. **D** Comparison of the posterior distributions of the infectivity reduction for NA and HA stalk titers. **E** Relative infectivity of children aged ≤15 years compared to adults aged >15 years (median and 95% CrI). The boxplots represent these five percentiles (2.5%, 25%, 50%, 75%, and 97.5%) of the posterior distributions. Orange designating neuraminidase (NA), purple designating the hemagglutinin (HA) head, and green designating the hemagglutinin (HA) stalk.

corresponding to the generation time for influenza, with differences in infectivity driven solely by variations in the amplitude of this distribution. However, differences in infectivity could also result from faster clearance of infectious virus and a reduction in the associated infectious period. We lacked the power to assess these potential differences in transmission dynamics. A potential future development could involve incorporating viral load data to estimate viral load trajectories and better inform infectivity based on pre-existing immunity.

This study focuses on the influenza A/H3N2 strain. To confirm the relevance of the tested antibodies in reducing transmission, particularly in the context of developing a future universal vaccine, the model could be extended to H1N1pdm transmission data. A previous publication has already demonstrated that antibodies against NA and HA are protective against infection[28]; however, their impact on case infectivity remains to be established.

In conclusion, using data from two large household transmission studies, we found that, though pre-existing anti-HA head, HA stalk, and anti-NA antibodies are important for reduced susceptibility to influenza A/H3N2 virus infection, only anti-NA and anti-HA stalk antibodies

are associated with reduced infectivity in a household transmission setting. Though most current-generation influenza vaccines include a neuraminidase component, the post-vaccination immune response to neuraminidase is generally poor, and anti-NA responses are not typically assessed as a part of seasonal vaccine immunogenicity evaluation[30]. Our results suggest that the induction of a better humoral immune response against NA may improve next-generation vaccines' effectiveness at preventing infection and disease, and may reduce individual infectivity even in the event of a breakthrough infection. These findings reinforce the need for continued development of influenza vaccinations that target NA in addition to HA in order to develop next-generation influenza vaccines that protect against influenza virus infection and reduce influenza infectivity.

## Methods

### Study population and design

This study uses data from two household influenza transmission studies based in Managua, Nicaragua: the Household Influenza Transmission Study (HITS) and the Household Influenza Cohort Study

(HICS). HITS is a case-ascertained study, meaning that influenza-positive individuals are identified, and other members of their household recruited for enrollment, that ran from 2012 to 2017, and HICS is a prospective household-based cohort study that began in 2017 and is currently ongoing. In both studies, influenza A/H3N2 virus-positive individuals, the index cases, are initially detected at a health center, where household members are enrolled (HITS) or activated (HICS) into intensive monitoring for a period of ~14 days. During this period, household members are tested repeatedly for influenza virus, allowing for a reconstruction of likely transmission chains within each household. Blood samples are collected both at the beginning of the monitoring period, no later than 7 days after symptom onset or household activation, and 30–45 days after[38]. Prior work using this population has focused on ascertaining correlation between various antibody responses and protection against infection. This analysis expands on that work by modeling protection against infection utilizing more sophisticated household transmission modeling, as well as modeling the association between antibody measurements and infectivity[28]. These studies were approved by the institutional review boards at the Nicaraguan Ministry of Health/Center National of Diagnosis and Reference Institutional Ethics Review Committee (Code NIC-MINSA/CNDR CIRE-05/04/17-080 Rev 13) and the University of Michigan (IRB Approval # HUM00119145) and are in accordance with the Helsinki Declaration of the World Medical Association. Written consent to participate or parental permission was obtained for all participants; in children older than 6 years, verbal assent was obtained. Data on gender were collected from participants using self-reports. Data were not stratified by gender for foundational models, given limited statistical power and the absence of compelling prior evidence that the association between antibody levels and infectivity differs by gender. Data were collected via trained study personnel conducting home visits utilizing tablets to import survey data. Data were collected on all individuals enrolled in the study who met criteria for an intensive monitoring period, so sample size calculations were not performed.

## Laboratory methods

Nasal/oropharyngeal swabs collected from household members were tested for influenza virus with real-time reverse-transcription polymerase chain reaction (RT-PCR) using validated Centers for Disease Control and Prevention (CDC) protocols. If positive for influenza virus, subtype or lineage determination was performed using additional RT-PCR assays[39–41]. Pre-transmission antibody levels were assessed by testing each sample using (1) HAI along with enzyme-linked immunosorbent assays (ELISAs) directed at antibodies against (2) full-length HA, the (3) HA stalk, and (4) NA. HAIs were performed to test the ability of participant antibodies to neutralize influenza virus HA's agglutination of turkey red blood cells, a proxy for antibody responses against the HA head. ELISAs were performed to test participant antibody responses against full-length HA, the HA stalk, and NA. The HAI of samples against A/Hong Kong/4801/2014 was tested. Immunoglobulin G (IgG) antibodies against trimeric H3 A/Hong Kong/4801/2014 and tetrameric N2 A/Hong Kong/4801/2014 were measured by ELISA using an anti-human IgG (Fab specific) horseradish peroxidase detector[42]. Antibody responses against the HA stalk were measured using a cHA (cH7/3) protein that expresses a head domain to which participants should be naive (A/Anhui/1/2013) along with the stalk of H3 (A/Hong Kong/4801/2014), ensuring that detected immune responses against the cHA are directed at only the HA stalk. The analyses utilized assay data from initial, baseline samples, which were collected either before or shortly after household activation, when anti-influenza IgG levels should approximate preexposure levels. Sera were treated with receptor-destroying enzyme and incubated overnight at 37 °C. Following inactivation and dilution in saline, 50 μL of inactivated sera was added to the first column of a 96-well V-bottom plate, and a two-fold serial dilution was performed across columns 1–10 (final dilutions ranging from 1:10 to 1:5120). Influenza virus antigen was added to columns 1–10, and 0.5% turkey red blood cells were then added to all wells. Hemagglutination inhibition was read after 30 min of incubation at room temperature. The HAI titer was defined as the reciprocal of the highest serum dilution that completely inhibited hemagglutination.

We used only RT-PCR assays to avoid potential bias that could arise from using serology, as individuals with high pre-existing antibody levels, which are the focus of this study, might have a higher probability of being classified as infected. In total, 113 of 516 (21.8%) PCR-negative individuals exhibited a fourfold or greater rise in HAI titers, possibly reflecting a host immune response to circulating household infection in individuals who do not develop an explicitly RT-PCR positive infection. These individuals would likely be classified as an infection under a less stringent definition, which reflects the imperfect specificity of HAI compared to RT-PCR as the gold standard. Several serological assays were conducted on each blood sample to measure the initial and final antibody levels against various influenza antigens; hemagglutination inhibition assays (HA heads), and enzyme-linked immunosorbent assays (ELISAs) against full-length HA, the HA stalk, and NA. Details about the specific antigens used for each assay are available in the supplement (Table D1).

Recombinant antigen proteins were diluted in 1× phosphate-buffered saline (PBS) to a final concentration of 2 μg/mL. A 50 μL volume of each antigen solution was added to 96-well microtiter plates (Immulon 4 HBX; Thermo Scientific, cat. no. 439454) and incubated overnight at 4 °C. Plates were washed three times with PBS-T (1× PBS containing 0.1% Tween 20) using an automated plate washer (BioTek 405TS). Blocking was performed with 200 μL/well of 3% (w/v) non-fat milk powder in PBS-T for 1 h at room temperature (RT). After removing the blocking buffer, heat-inactivated serum samples (mouse or human) were serially diluted in 1% (w/v) milk-PBS-T, starting at a 1:100 dilution, followed by twofold dilution steps and incubated for 2 h at RT. Plates were subsequently washed three times with PBS-T.

Anti-human IgG (Fab-specific)-HRP (Sigma-Aldrich, A0293; 1:3000) diluted in 1% milk-PBS-T was added (50 μL/well) and incubated for 1 h at RT. After plates were washed three times with PBS-T, 100 μL/well of o-phenylenediamine dihydrochloride (OPD; SIGMAFAST) substrate was added. The reaction was stopped after 10 min with 50 μL/well of 3 M HCl (ThermoFisher). Optical density (OD) at 490 nm was measured using a BioTek Synergy H1 or Synergy 4 plate reader.

Area under the curve (AUC) was calculated to quantify total antibody binding across serial dilutions. OD values at each dilution were plotted against dilution factor, and the AUC was determined by integrating the curve using GraphPad Prism software."

## Statistical methods

Descriptive statistics of the study population are presented in Table 1. For categorical variables (e.g., gender, vaccination status, season), the number of individuals and their proportions are reported for each category. $P$ values are calculated using a chi-squared test of independence to determine whether the proportions significantly differ between the infected and noninfected groups.

For continuous variables (e.g., age, antibody levels), we report the median and standard deviation within each group (infected and noninfected). The Mann–Whitney U test is used to assess whether the distributions of these variables differ significantly between the two groups, as this test does not assume a parametric distribution. This test is robust to left-censoring of observations, as it relies on the relative ranking of the data rather than their absolute values. $P$ values < 0.05 are considered statistically significant, indicating a difference in distributions at a 5% risk level.

We performed a first regression analysis using boosted regression trees to assess the association between the age and antibody titers of the index case and household contacts and the probability of infection in the household contacts. Boosted regression trees were chosen

because they can capture non-linear relationships in the data and account for potential interactions between variables[43–45].

The XGBoost algorithm was used to partition the co-variate space defined by antibody levels and age and to assign each subspace a corresponding probability of infection. To minimize the risk of over-fitting, we implemented 5-fold cross-validation[46]. The optimal number of boosting rounds (iterations) was selected using early stopping, ensuring the model stops training once further iterations do not improve performance on the validation set.

We present the estimated association of each covariate with the probability of infection, along with confidence intervals derived using a bootstrap procedure. Specifically, we resampled 80% of the training set 500 times and re-estimated the model parameters for each resampling.

To evaluate global co-variate importance, we used the mean absolute SHapley Additive exPlanations (SHAP). SHAP values measure the average contribution of each covariate to the prediction[47]. For a given observation $i$, the SHAP value $i$, $\phi_{ij}$ for covariate $j$ represents the marginal contribution of that covariate, averaged across all possible combinations of covariates. The mean absolute SHAP value quantifies the overall importance of a covariate while ignoring the directionality (sign) of its effect.

$$\text{Mean Absolute SHAP}_j = \frac{1}{n}\sum_{i=1}^{n}\left|\phi_{ij}\right|, \; j \; one \; feature \; and \; i \; one \; observation \quad (1)$$

Finally, we computed the $p$-value associated with each variable using a permutation test[48]. To perform this test, we randomly permuted the observations and computed the mean absolute SHAP value for each permutation. We then compared the proportion of permutations in which the mean absolute SHAP value exceeded the real SHAP value. If the observed SHAP value is consistently larger than the permuted SHAP values, it indicates that the covariate has significant importance in predicting the outcome beyond what would be expected by chance.

$$p-value = \frac{N\{permutation \; SHAP > observed \; SHAP\}}{N\{permutation\}} \quad (2)$$

Analyses were conducted using R version 4.2-4.5.0 and Visual Studio Code version 1.82.0-1.104.

## Transmission model and analysis

We used a mathematical model to assess the impact of individual-level age and immune characteristics on the person-to-person probability of transmission. The model estimated the risk of transmission between all household members, including the risk from secondary cases. We modeled how the risk of transmission from an infected individual varied according to time after infection with a gamma distribution[24,49–51] with a median at 3.5 days and a standard deviation of 2 days. This transmission risk was modulated by infectivity factors, including the infected individual's pre-existing antibody titers against HA head, HA stalk, and NA. It was further influenced by susceptibility factors, which comprised the characteristics of the susceptible contact, namely their age and pre-existing antibody titers (anti-HA head, anti-HA stalk, and anti-NA). Household size was also included as a covariate. Additionally, we estimated a baseline risk of infection from the community. Compared to traditional approaches, such as logistic models[24,31,51,52], a key advantage of our transmission model is that it can simultaneously account for transmission risks from the index case, other infected household members, and external community sources. Full details of the transmission model are provided in the Supplementary Information.

We estimated how transmission risk was modulated by individual antibody titers. For each antibody type, we estimated two pairs of parameters: one pair quantifying the reduction in infectivity and

another pair quantifying the reduction in susceptibility. The first parameter represented the threshold antibody level at which infectivity or susceptibility was reduced, and the second quantified the associated reduction in transmission. Reductions in susceptibility, reductions in infectivity, and their corresponding antibody thresholds were jointly estimated within the transmission model.

## Inference framework

Inference is complicated by the fact that times of infection are not observed. We used a Bayesian data augmentation approach to address this missing data issue[24,53]. In this framework, unobserved times of infection are considered as "augmented data" and the joint posterior distribution of transmission parameters and augmented data is explored by Markov chain Monte Carlo (MCMC). The statistical model had a hierarchical structure with three levels: (i) the observation level ensures consistency between observed and augmented data based on the probabilistic distribution assumed for the incubation period[54–56]. The incubation period is modeled as a log-normal distribution with a median of 1 day and a standard deviation of 1.2 days. (ii) the transmission model (described above), characterizes within household transmission dynamics, (iii) the prior model describes prior distribution for model parameters. They are reported in the "Priors of the Reference Transmission Model" section of the Supplementary Information. Transmission parameters and augmented times of infection were iteratively updated using a Metropolis-Hastings algorithm [24,53,57]. Each MCMC chain was run for 50,000 iterations, with the first 10,000 iterations discarded as burn-in. We report the median of the posterior distributions along with the 95% credible intervals (CrI) for each estimated parameter. Convergence was assessed using the Gelman-Rubin diagnostic ($\hat{R}$) and reported the values in Table S15.

The prior for relative risks was specified as a log-normal distribution with a log-mean of 0 and a log-standard deviation (log-SD) of 1. The baseline threshold priors were informed by visual inspection of results from boosted regression trees. For susceptibility, the threshold prior was specified as a uniform distribution from 10 to 70. For infectivity, the NA threshold was assigned a uniform prior from 10 to 50, and the HA stalk threshold from 40 to 70. More details are reported in the supplement.

## Sensitivity analysis

In our baseline model, we assumed that the effects of different types of antibodies on susceptibility and infectivity were additive. In a sensitivity analysis, we assessed this assumption by evaluating the combined effects of high antibody levels across multiple antigens. Using the thresholds estimated from the baseline model, we categorized individuals as having high or low antibody levels for infectivity and susceptibility analyses. For susceptibility, we evaluated the effect of: (i) a single high antibody level; joint high levels of two antibodies (HA head and HA stalk; NA and HA stalk; NA and HA head); combined high levels of all three antibodies. For infectivity, we excluded HA head due to its lack of association with reduced infectivity and assessed the reduction associated with high levels of NA only, HA stalk only, or their joint high levels.

We performed sensitivity analyses on prior assumptions related to viral life history traits, including the incubation period and generation time distributions. In the baseline model, we assumed a generation time with a mean of 3.5 days and a standard deviation of 2 days. Alternative scenarios included a shorter generation time (mean = 3 days, SD = 1.5 days) and a longer one (mean = 4 days, SD = 2 days). The baseline incubation period was set to a mean of 1 day with a standard deviation of 1.2 days. We also tested a shorter incubation period (mean = 0.6 days, SD = 0.8) and a longer one (mean = 2 days, SD = 1.6).

In a sensitivity analysis, we tested a wider prior with a log-SD of 2 for relative risk parameters. We also performed sensitivity analyses on

the priors for the antibody titer activation thresholds. We tested uniform priors centered around the target thresholds with varying widths of 20, 40, 60, and 80. For example, with a width of 20, the priors were set to 25–45 for susceptibility, 20–40 for NA infectivity, and 50–70 for HA stalk infectivity.

## Model validation and simulation

To validate the model, we generated 2000 datasets of infection events using an agent-based model. The simulations preserved the household structure, index case assignment, and individual characteristics such as age and antibody levels. At each time step, infection events were randomly sampled based on the probability of transmission derived from the transmission model, using the posterior medians of the estimated transmission parameters.

Model adequacy was assessed by comparing the secondary attack rates (SARs) across different household sizes between the simulated datasets and the observed data. To evaluate parameter identifiability, we calculated the proportion of the posterior distribution that included the simulation-derived parameter values within the 95% credible interval (CrI). A proportion close to 95% ensures that the parameter is well identifiable.

All analyses were performed, R (versions 4.3.1–4.3.2), Visual Studio Code (version 1.87.2) and Python 3.13.0 for the XGBoost analysis.

## Reporting summary

Further information on research design is available in the Nature Portfolio Reporting Summary linked to this article.

## Data availability

Researchers interested in accessing the study data are encouraged to submit a formal request to A.G. or the Health Sciences and Behavioral Sciences Institutional Review Board at the University of Michigan. To uphold ethical standards and ensure appropriate data use, each request will undergo a case-by-case review and approval process. Additionally, as the data include information collected in Nicaragua, access is subject to Nicaraguan data ownership regulations and may require approval from relevant Nicaraguan authorities. Final approval is expected within two months, and if access is granted, the duration of data access will be granted on a case-by-case basis.

## Code availability

The code that support the findings of this study are available for download and review here: https://gitlab.pasteur.fr/mmmi-pasteur/flu-antibody-model.

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

## Acknowledgements

This work was supported by the National Institutes of Allergy and Infectious Diseases through the Collaborative Influenza Vaccine Innovation Centers [75N93019C00051 to F.K. and A.G.], the St. Jude Center of Influenza Research and Surveillance [HHSN272201400006C to A.G.]; the St. Jude Center of Excellence for Influenza Research and Response [75N93021C00016 to A.G.] and grant R01 AI120997 to A.G. A.G. is supported by the Biosciences Initiative at the University of Michigan through a Mid-career Biosciences Faculty Achievement Award (MBio-FAR). S.C. acknowledges support by the European Commission under the EU4Health programme 2021–2027, Grant Agreement–Project: 101102733–DURABLE, the Laboratoire d'Excellence Integrative Biology of Emerging Infectious Diseases program (grant ANR-10-LABX-62-IBEID) the European Union's Horizon 2020 research and innovation programme under VEO grant agreement No. 874735, AXARF and the INCEPTION project (PIA/ANR16-CONV-0005). The funding agencies had no role in the design and conduct of the study, collection, management, analysis or interpretation of the data; preparation, review, or approval of the manuscript; or decision to submit the manuscript for publication. We thank the study participants and the many dedicated study personnel in Nicaragua at the Centro Nacional de Diagnóstico y Referencia and the Sócrates Flores Vivas Health Center.

## Author contributions

Data curation: Conception and design: S.C, A.G. Data acquisition and provision: G.H., T.C., H.M., D.S., and A.S. Analysis and interpretation: G.H., T.C., H.M., S.C., A.G. Investigation and project administration: G.K., R.L, N.S., S.O., M.P. Manuscript drafting: G.H., T.C., H.M., S.C., A.G. Manuscript revision: G.H., T.C., H.M., G.K., R.L., N.S., S.O., M.P., D.S., A.S., A.B., F.K., S.C., A.G. Supervision: A.B., S.C., and A.G.

## Competing interests

The Icahn School of Medicine at Mount Sinai has filed patent applications related to SARS-CoV-2 serological assays, NDV-based SARS-CoV-2 vaccines, influenza virus vaccines and influenza virus therapeutics, which list Florian Krammer as co-inventor. Mount Sinai has spun out a company, Kantaro, to market serological tests for SARS-CoV-2 and another company, CastleVax, to develop SARS-CoV-2 vaccines. Florian Krammer is a co-founder and scientific advisory board member of CastleVax. Florian Krammer has consulted for Merck, CureVac, Seqirus, GSK and Pfizer and is currently consulting for 3rd Rock Ventures, Gritstone and Avimex. The Krammer laboratory is collaborating with Dynavax on influenza vaccine development. Aubree Gordon has served on an RSV vaccine advisory board for Janssen. All other authors declare no competing interests.
