## [Peer Review file · Nature Communications]

Anti-Neuraminidase and anti-HA stalk Antibodies Reduce the Susceptibility to and Infectivity of Influenza A/H3N2 Virus

Corresponding Author: Professor Aubree Gordon

Version 0:

Reviewer comments:

Reviewer #2

(Remarks to the Author)

Thank you for the opportunity to review the revised manuscript. The authors have made substantial improvements that address prior comments. The use of boosted regression trees (BRTs) and the presentation of results are clear and analytically appropriate. The revised manuscript reports reduced susceptibility associated with anti-HA head, anti-HA stalk, and anti-NA antibodies, and reduced infectivity associated with anti-HA stalk and anti-NA—but not anti-HA head—antibodies. The BRT results also suggest nonlinear relationships between these antibody measures and both susceptibility and infectivity. These findings are important for understanding how pre-existing immunity shapes risk of A(H3N2) infection. I only have two minor suggestions remaining.

1. Figure 2B - A sizeable proportion of observations appear below the detection threshold across groups. It would help to clarify which statistical tests were used to compare children vs. adults and how left-censoring/excess zeros were handled. Also, for the p-value labels in the top-right and bottom-middle panels, I suppose the authors intended to label as “ $p < 2e-16$ ”?

2. Figure 5 - Since the HA head infectivity estimate is also discussed in the main text and Figure 4, the authors might consider presenting all three antibody classes (HA head, HA stalk, NA) together in Figure 5 for completeness, even if some effects are non-significant. The current results shown in Supplementary Figure 1 would also work, but moving the results to the main text may help readers compare magnitudes and CIs at a glance.

(Remarks on code availability)

Reviewer #3

(Remarks to the Author)

The authors have done a great job addressing all reviewer comments from the first round of review. I have a few remaining minor questions and suggestions, but otherwise recommend publication with minimal revisions.

Remaining comments (minor)

- The main methods states “For each antibody type, we estimated one parameter to quantify the reduction in infectivity and one parameter to quantify the reduction in susceptibility.” But in the supplementary methods it shows there are two parameters (one for the threshold and one for the risk reduction). Suggest updating the main methods.
- “We used only RT-PCR assays to avoid potential bias that could arise from using serology, as individuals with high pre-existing antibody levels, which are the focus of this study, might have a higher probability of being classified as infected.” That makes sense, but I would suggest reporting how many likely serology-based infections are being excluded under this criteria.
- The priors on the antibody activation thresholds seem quite tight. I appreciate the sensitivity analyses around these, but I

wonder if the authors could justify these tight bounds? The prior on the scaling factors seem appropriately wide so I am not concerned about the validity of the estimated protective effect. It might be helpful to include a prior predictive plot with the same layout to Figure 4.

- "Given the uncertainty surrounding the couple of threshold and susceptibility/infectivity reduction values, our results effectively trace out a form of dose-response curve." Just a pedantic point, but this is only true visually – each posterior draw uses the step function. There is no realised projection of the model which uses a dose-response curve.

- Figure 1: I appreciate the inclusion of this figure, but I don't think it is totally clear. That's my subjective opinion so not a required revision if the authors are happy with the figure. The insets for intracellular mechanisms are unlabelled (antibodies vs. virions) and don't really make it clear how the mathematical model works. For example, I might replace the central equation with "Probability of infection = case infectivity x contact susceptibility x other factors" or similar, and have arrows from the case infectivity/contact susceptibility to the different potential factors. Furthermore, why use the term "NA load" rather than NA titer? I'd also suggest expanding the caption.

- Figure 3: I appreciate including this additional analysis. Personally I think the transmission model alone was sufficient, but I understand that some readers are more trusting of "standard" statistical models. Would it be better to show *relative* probability of being infected/infecting compared to baseline for each covariate? The caption is also a bit unclear, maybe a slight grammar issue e.g., should it read "The first row shows the effect of a contact's individual factors on the probability of becoming infected. The second row shows the effect of an index case's factors on their probability of infecting the contact."?

- Figure 4: in D/E, the caption states "comparison of posterior distributions of the threshold values", but these are boxplots – which summary statistics do they represent?

- Table S4: the % of CrI covering the true simulated value seems high across the board, but I am not sure what the ref% for Ri_NA means? I think this is probably fine, but I just want to check that the authors are not calculating the % of CrI containing the true value relative to the % of Ri_NA CrIs containing the true value. That would be odd, but I can't think what ref% means otherwise.

- Figure S1: this is a helpful figure, but what are the true simulated parameter values? I suggest marking these so that we can interpret the estimates alongside the true values.

- Figure S2: what does Secondary attack rate in actual and simulated datasets refer to? Are these combined results from both real and simulated data? I'm not sure why you would present both together.

James Hay

(Remarks on code availability)

I looked through the GitHub repository and attempted to rerun the code. The code is neat and well commented, but I was not able to rerun any of the analyses.

For the real data analyses, I understand that I cannot run the real data analyses as the data cannot be made publicly available. It looks like the main script requires a cluster, and thus it's unclear how one would run the model locally.

For the simulation analyses, the main script is missing the file "flu_index_simulation.txt" which stops the script. It also looks like the number of arguments for hhEpidemic is 12 in the cpp file, but 18 where it's called in the R script. I cannot run the function even with adding dummy values for sim_database_index. I did not check the code past this point as I would need the output of the hhEpidemic function to continue. I would request that the authors make sure the simulation model can be run, including fitting the model with MCMC, and ideally provide a more comprehensive readme so that others can reproduce the full pipeline. Also, the authors have ticked "code availability statement made available" in the reporting checklist but I cannot see this in the manuscript.

I would suggest including a more comprehensive readme file on the landing page for the full repo, as the details are minimal. They don't, for example, describe the workflow, pre-requisites, data, package dependencies etc.

Reviewer #2 (Remarks to the Author):

Thank you for the opportunity to review the revised manuscript. The authors have made substantial improvements that address prior comments. The use of boosted regression trees (BRTs) and the presentation of results are clear and analytically appropriate. The revised manuscript reports reduced susceptibility associated with anti-HA head, anti-HA stalk, and anti-NA antibodies, and reduced infectivity associated with anti-HA stalk and anti-NA—but not anti-HA head—antibodies. The BRT results also suggest nonlinear relationships between these antibody measures and both susceptibility and infectivity. These findings are important for understanding how pre-existing immunity shapes risk of A(H3N2) infection. I only have two minor suggestions remaining.

We thank the reviewer for their positive feedback.

1. Figure 2B - A sizeable proportion of observations appear below the detection threshold across groups. It would help to clarify which statistical tests were used to compare children vs. adults and how left-censoring/excess zeros were handled. Also, for the p-value labels in the top-right and bottom-middle panels, I suppose the authors intended to label as “ $p < 2e-16$ ”?

Response: Thank you for the remark. As mentioned in the Methods, the test used is a Mann–Whitney U test, which is based on the ranking of the data rather than their absolute values, making it robust to left-censoring. We have added the following sentence to the Methods: ‘This test is robust to left-censoring of observations, as it relies on the relative ranking of the data rather than their absolute values.’ For the p-value, we have corrected the label by removing the “=”.

2. Figure 5 - Since the HA head infectivity estimate is also discussed in the main text and Figure 4, the authors might consider presenting all three antibody classes (HA head, HA stalk, NA) together in Figure 5 for completeness, even if some effects are non-significant. The current results shown in Supplementary Figure 1 would also work, but moving the results to the main text may help readers compare magnitudes and CIs at a glance.

Response: After careful consideration, we decided to present only the most parsimonious version of the model in the main text, including only the significant effects to provide the most precise estimates. A version with all effects is provided in the supplementary analysis.

Reviewer #3 (Remarks to the Author):

The authors have done a great job addressing all reviewer comments from the first round of review. I have a few remaining minor questions and suggestions, but otherwise recommend publication with minimal revisions.

We thank the referees for their positive feedback.

Remaining comments (minor)

- The main methods states “For each antibody type, we estimated one parameter to quantify the reduction in infectivity and one parameter to quantify the reduction in susceptibility.” But in the supplementary methods it shows there are two parameters (one for the threshold and one for the risk reduction). Suggest updating the main methods.

Response: Thank you. To improve coherence, we revised the main Methods section as follows: “We estimated how transmission risk was modulated by individual antibody titers. For each antibody type, we estimated two pairs of parameters: one pair quantifying the reduction in infectivity and another pair quantifying the reduction in susceptibility. The first parameter represented the threshold antibody level at which infectivity or susceptibility was reduced, and the second quantified the associated reduction in transmission. Reductions in susceptibility, reductions in infectivity, and their corresponding antibody thresholds were jointly estimated within the transmission model.”

- “We used only RT-PCR assays to avoid potential bias that could arise from using serology, as individuals with high pre-existing antibody levels, which are the focus of this study, might have a higher probability of being classified as infected.” That makes sense, but I would suggest reporting how many likely serology-based infections are being excluded under this criteria.

Response: We have reported the number of infections that would be called using a standard 4-fold rise approach compared to the gold-standard RT-PCR approach used in this study. The main text now reads “In total, 113 of 516 (21.8%) PCR-negative individuals exhibited a 4-fold or greater rise in HAI titers, possibly reflecting

a host immune response to circulating household infection in individuals who do not develop an explicitly RT-PCR positive infection. These individuals would likely be classified as an infection under a less stringent definition, which reflects the imperfect specificity of HAI compared to RT-PCR as the gold standard.”

- The priors on the antibody activation thresholds seem quite tight. I appreciate the sensitivity analyses around these, but I wonder if the authors could justify these tight bounds? The prior on the scaling factors seem appropriately wide so I am not concerned about the validity of the estimated protective effect. It might be helpful to include a prior predictive plot with the same layout to Figure 4.

Response: The priors on antibody threshold activation are informed by the boosted regression trees model. The main challenge for the transmission model was to remain within a biologically plausible range when estimating the optimal threshold across multiple antibodies. In the sensitivity analysis, we examined the impact of varying the width of the prior window. We found that for wider windows (60 and 80), the estimated reduction in transmission due to lower infectivity was only borderline significant, although the median estimate of the reduction effect remained stable. For clarity, we chose to present the scenario with informed priors in the main text. To emphasize this, we added the following when introducing the sensitivity analysis: *‘We present a scenario with an informed prior in the main text for the sake of clarity.’*

- “Given the uncertainty surrounding the couple of threshold and susceptibility/infectivity reduction values, our results effectively trace out a form of dose–response curve.” Just a pedantic point, but this is only true visually – each posterior draw uses the step function. There is no realised projection of the model which uses a dose-response curve.

Response: The referee is correct, the underlying model is still a step model, but we thought it could provide an idea of the curve’s shape that we would obtain if we had the power to run the inference with a continuous model.

- Figure 1: I appreciate the inclusion of this figure, but I don’t think it is totally clear. That’s my subjective opinion so not a required revision if the authors are happy with the figure. The insets for intracellular mechanisms are unlabelled (antibodies vs. virions) and don’t really make it clear how the mathematical model works. For example, I might replace the central equation with “Probability of infection = case infectivity x contact susceptibility x other factors” or similar, and have arrows from the case infectivity/contact

susceptibility to the different potential factors. Furthermore, why use the term “NA load” rather than NA titer? I'd also suggest expanding the caption.

Response : Thanks for the suggestions. We have clarified the figure by making it clearer what the antibodies were. We have slightly increased the size of the legends and clarified the mathematical translation at the top. The version shown in the figure was correct, with the risk of transmission being proportional to case infectivity and contact susceptibility. We wanted to keep only these two factors on the figure, as they represent the main message of the paper. We have also extended the caption, which is now : Figure 1. Factors influencing pairwise transmission risk, as tested across the different models used in the study. Zoomed bubbles strands for the potential cellular mechanism of antibodies on modulation of infectivity (left) and susceptibility (right). NA refers to neuraminidase, and HA refers to hemagglutinin. Created in BioRender. Cortier, T. (2025) <https://BioRender.com/tlkiwj8>

- Figure 3: I appreciate including this additional analysis. Personally I think the transmission model alone was sufficient, but I understand that some readers are more trusting of "standard" statistical models. Would it be better to show *relative* probability of being infected/infecting compared to baseline for each covariate? The caption is also a bit unclear, maybe a slight grammar issue e.g., should it read “The first row shows the effect of a contact’s individual factors on the probability of becoming infected. The second row shows the effect of an index case’s factors on their probability of infecting the contact.”?

Response : Thank you for the suggestion to improve clarity. We’ve modified the caption to make it grammatically correct.

- Figure 4: in D/E, the caption states “comparison of posterior distributions of the threshold values”, but these are boxplots – which summary statistics do they represent?

Response : Thank you for noticing the lack of precision. The boxplots represent the 2.5%, 25%, 50%, 75%, and 97.5% percentiles. We’ve added the following sentence to the captions of both Figures 4 and 5: ‘The boxplots represent these five percentiles (2.5%, 25%, 50%, 75%, and 97.5%) of the posterior distributions.’

- Table S4: the % of CrI covering the true simulated value seems high across the board, but I am not sure what the ref% for Ri_NA means? I think this is probably fine, but I just want to check that the authors are not calculating the

% of CrI containing the true value relative to the % of Ri_NA CrIs containing the true value. That would be odd, but I can't think what ref% means otherwise.

Response: Thank you for the comment. Indeed, we do calculate the percentage of the CrI that contains the simulation value. The reference was a typo, which has been corrected in the latest version of the Supplementary Materials.

- Figure S1: this is a helpful figure, but what are the true simulated parameter values? I suggest marking these so that we can interpret the estimates alongside the true values.

Response: Thanks for spotting that mistake. The beginning of the legend of this supplementary figure was an unfortunate copy-paste and is unrelated to the simulated parameters. It simply presents the complete model, including the estimate of the effect of the HA head on infectivity. It is now corrected.

- Figure S2: what does Secondary attack rate in actual and simulated datasets refer to? Are these combined results from both real and simulated data? I'm not sure why you would present both together.

Response: Again, thank you for spotting that mistake. The beginning of the caption was again an unfortunate copy-paste and is unrelated to the secondary attack rate. It shows the posterior distribution of the relative transmission risk across influenza waves and across household sizes. I have corrected the beginning of the caption to: 'Temporal and demographic variation in transmission risk.'

James Hay

Reviewer #3 (Remarks on code availability):

I looked through the GitHub repository and attempted to rerun the code. The code is neat and well commented, but I was not able to rerun any of the analyses.

For the real data analyses, I understand that I cannot run the real data analyses as the data cannot be made publicly available. It looks like the main script requires a cluster, and thus it's unclear how one would run the model locally.

For the simulation analyses, the main script is missing the file

"flu_index_simulation.txt" which stops the script. It also looks like the number of arguments for hhEpidemic is 12 in the cpp file, but 18 where it's called in the R script. I cannot run the function even with adding dummy avlues for sim_database_index. I did not check the code past this point as I would need the output of the hhEpidemic function to continue. I would request that the authors make sure the simulation model can be run, including fitting the model with MCMC, and ideally provide a more comprehensive readme so that others can reproduce the full pipeline. Also, the authors have ticked "code availability statement made available" in the reporting checklist but I cannot see this in the manuscript.

I would suggest including a more comprehensive readme file on the landing page for the full repo, as the details are minimal. They don't, for example, describe the workflow, pre-requisites, data, package dependencies etc.

Response: We apologize for the initial version of the repository, which did not meet the standards expected for a code availability statement. We have made the required improvements and updates. Readers should now be able to run the previously missing simulation code and perform inference based on simulated data. The README has been updated with a more detailed procedure for running the inference code, and the simulation code and usage instructions have been revised. We hope this updated repository will enable readers to more easily reproduce and explore the analyses presented in the paper.